

# Effects of origin, seasons and storage under different temperatures on germination of *Senecio vulgaris* (Asteraceae) seeds

Noel Ndihokubwayo[1,2,*], Viet-Thang Nguyen[1,3,*] and Dandan Cheng[4]

[1] School of Environmental Studies, China University of Geosciences (Wuhan), Wuhan, China
[2] Département des Sciences Naturelles, Ecole Normale Supérieure, Bujumbura, Burundi
[3] Faculty of Biology, Thai Nguyen University of Education, Thai Nguyen City, Vietnam
[4] State Key Laboratory of Biogeology and Environmental Geology, China University of Geosciences (Wuhan), Wuhan, China
[*] These authors contributed equally to this work.

## ABSTRACT

Invasive plants colonize new environments, become pests and cause biodiversity loss, economic loss and health damage. *Senecio vulgaris* L. (Common groundsel, Asteraceae), a widely distributing cosmopolitan weed in the temperate area, is reported with large populations in the north–eastern and south–western part, but not in southern, central, or north-western parts of China. We studied the germination behavior of *S. vulgaris* to explain the distribution and the biological invasion of this species in China. We used seeds originating from six native and six invasive populations to conduct germination experiments in a climate chamber and under outdoor condition. When incubated in a climate chamber (15 °C), seeds from the majority of the populations showed >90% germination percentage (GP) and the GP was equal for seeds with a native and invasive origin. The mean germination time (MGT) was significantly different among the populations. Under outdoor conditions, significant effects of origin, storage conditions (stored at 4 °C or ambient room temperature, ca. 27 °C) and seasons (in summer or autumn) were observed on the GP while the MGT was only affected by the season. In autumn, the GP (38.6%) was higher and the MGT was slightly longer than that in summer. In autumn, seeds stored at 4 °C showed higher GP than those stored at ambient room temperature (ca.27 °C), and seeds from invasive populations revealed higher GP than those from native populations. The results implied that the high temperature in summer has a negative impact on the germination and might cause viability loss or secondary dormancy to *S. vulgaris* seeds. Our study offers a clue to exploring what factor limits the distribution of *S. vulgaris* in China by explaining why, in the cities in South-East China and central China such as Wuhan, *S. vulgaris* cannot establish natural and viable populations.

Corresponding author
Dandan Cheng,
dandan.cheng@cug.edu.cn,
dan-d-cheng@163.com

## INTRODUCTION

Invasive plants colonize new areas, become pests and cause biodiversity loss, economic loss and health damage (*Keller et al., 2011*). An invasive species is a non-native species whose introduction does or is likely to cause economic or environmental harm or harm to human, animal, or plant health (*Horan & Lupi, 2010*). One of such invasive species is *Senecio vulgaris* (Common groundsel, Asteraceae) which most probably originated from southern Europe and is widely distributed in temperate areas all over the world (*Robinson et al., 2003*). Despite the wide distribution of *S. vulgaris* in China, its occurrence is scattered, with large populations reported in the north–eastern and south–western parts, but not in southern, central, northern or north-western parts of China (*Cheng & Xu, 2015*).

Germination is an important stage in the life cycle of plants, and germination behavior limits distribution. We germinated the seeds in autumn and summer in Wuhan, Central China, where no natural *S. vulgaris* populations are established. In Wuhan, we observed that the plants from seeds germinated in spring ended their life cycle in late spring or early summer with seed dispersion. Plants from seeds germinated in autumn grew in winter and ended in late spring as well. From this observation, we had a preliminary hypothesis that the *S. vulgaris* seeds cannot germinate or survive in the hot summer in Wuhan. This is the reason why no observable natural *S. vulgaris* populations established in Wuhan and other areas in the north–eastern and south–western parts of China where it is extremely hot in summer.

To test the hypothesis, we collected seeds from various populations from the native range (Europe) and invasive range (China), stored the seeds in different conditions (4 °C and ambient room temperature, ca. 27 °C), and germinated them in a controlled condition (15 °C) and outdoor conditions in summer and autumn in Wuhan. In particular, we addressed the following questions: (1) Do the storage conditions and seasons have an effect on seed germination; and (2) Does the germination behavior vary depending on the origin of range and the populations?

## MATERIALS AND METHODS

### Species description

*Senecio vulgaris* is an erect herbaceous annual plant growing up to 45 cm tall (*Stace, 1997*), has a thick taproot, and possesses an ephemeral strategy typical of many weedy species (*Weiner et al., 2009*). *S. vulgaris* is a ubiquitous weed found in the temperate zones of Europe, North and South America, North Africa and Asia (*Robinson et al., 2003*). In warmer climates such as California, however, it is a winter annual that appears soon after precipitation. Its optimal growing temperature is estimated to be 22 °C from meristem tips grown in static tube culture (*Walkey & Cooper, 1976*). Plants of *S. vulgaris* develop from seeds annually, and each plant can produce an average of 830 seeds (*Kadereit, 1984*). But, large plants of *S. vulgaris* can produce over 1,700 seeds (*Royer & Dickinson, 1999*).

### Seeds source

Seeds from nine populations of *S. vulgaris* in their native area (Europe) and seven populations in invasive areas (China) were sampled in different sites in 2012 and 2013,
**Table 1** Origin of the populations of *Senecio vulgaris* used in this study.

| Range | Country | Location | Collected time | Latitude | Longitude |
|---|---|---|---|---|---|
| Native | Spain | Barcelona[b] | June, 2012 | 41.67 | 2.73 |
| Native | Switzerland | Fribourg[b] | July, 2012 | 46.79 | 7.15 |
| Native | The Netherlands | Leiden[a,b] | Oct, 2013 | 52.17 | 4.48 |
| Native | The Netherlands | Lisse[a] | May, 2012 | 52.25 | 4.55 |
| Native | The Netherlands | Oegstgeest[a,b] | Oct, 2013 | 52.11 | 4.28 |
| Native | Germany | Potsdam[a] | July, 2012 | 52.39 | 13.06 |
| Native | Poland | Puławy[b] | July, 2012 | 51.39 | 21.96 |
| Native | United Kingdom | St Andrew[a,b] | May, 2012 | 56.33 | −2.78 |
| Native | The Netherlands | Teylingen[a] | Oct, 2013 | 52.21 | 4.49 |
| Invasive | China | Fuyuan[a,b] | July, 2013 | 48.37 | 134.29 |
| Invasive | China | Hegang[a,b] | July, 2013 | 47.33 | 130.29 |
| Invasive | China | Lijiang[a,b] | Sept, 2013 | 26.87 | 100.24 |
| Invasive | China | Luobei[a] | July, 2013 | 47.57 | 130.82 |
| Invasive | China | Siping[a,b] | July, 2013 | 43.17 | 124.38 |
| Invasive | China | Tongjiang[a,b] | July, 2013 | 47.98 | 133.17 |
| Invasive | China | Yichun[b] | July, 2013 | 47.72 | 128.79 |

**Notes.**
[a] Populations used in the germination experiment in a climate chamber.
[b] Populations used in outdoor germination experiments.

respectively (Table 1). After collection from the field, seeds were kept in paper bags and dried in air. Seed collection occurred between end of May and beginning of October, with most of the seeds collected in June and July (Table 1). Thus, most of the seeds were collected in summer. The plants from which seeds were collected were at least 5 m from each other within the same population. To avoid the maternal effect, we did not use seeds collected directly from the field.

Seeds of each population were grown for one generation in a climate room (20 °C, 18/6 h, light/dark) in October and November 2013. One set of seeds collected from these plants grown in climate room was used for the germination experiment in a climate chamber (PQX-1000A-12HM; Ningbo Southeast Instrument Co. Ltd, Ningbo, China) in December 2013 and January 2014 at 3–4 weeks later after seeds harvesting.

Another set of seeds was germinated and then cultivated in a greenhouse in spring 2014 and harvested in June of the same year. The resulting seeds of the first flowering capitulum were harvested and used in the outdoor germination experiments. From each population, three individual plants that contained a large number of good seeds were selected.

## Germination experiment design
### Germination in a climate chamber
From the 16 populations, we used six native and six invasive populations for the germination experiment in a climate chamber. Three replications (plants) per population, 10 seeds per plant, and in total 360 seeds were used for this experiment. These seeds did not experience any pretreatment, only were air-dried in paper bags and kept in room temperature in winter (below 20 °C).
The single layer of Whatman No.1 filter paper was placed inside a Petri dish (9 cm diameter) and moistened with distilled water. Ten seeds from the same plant were sown on a filter paper. The Petri dishes containing seeds were then placed in a climate chamber (Ningbo Southeast Instrument Co., Ltd, Ningbo, China). According to previous work, we selected good conditions for *S. vulgaris* seed to germinate: temperature was 15 °C, and 12 h for light (provided by 12 fluorescent tubes) and 12 h for darkness. Germinated seeds were recorded daily and the test criterion was the protrusion of the radicle. The data collection continued until germination had ceased after 19 days.

### *Outdoor germination experiments*

For these experiments, three plants from each of the six native and six invasive populations were selected, and 40 seeds were chosen from the same individual plant. These 40 selected seeds were kept in two paper bags, 20 seeds in each one. These bags were divided into two lots and each lot consisted of seeds collected from 36 plants representing 12 populations. The lots were then stored under two different temperature conditions: (i) ambient room temperature: seeds in the paper bag were placed in plastic bag containing a bag of silica gel to absorb moisture thereby abating humidity inside the plastic bag and placed in a cardboard box at ambient conditions in the laboratory. The temperature in the laboratory ranged from 20 to 30 °C and relative humidity was around 70% during the storage period; (ii) At low temperature: Another lot of seeds was put in plastic bag and tightly sealed and stored in a refrigerator (4 °C). In total 1,440 seeds were used in this experiment. The germination experiment was carried out twice, in July and in October. The experiment done in July used seeds stored for one month (seeds harvested in June) while the experiment conducted in October used seeds stored for four months.

In July (summer), 10 seeds per plant from lots stored at different conditions were sown on a filter paper soaked with tap water in Petri dishes. After sowing, Petri dishes were placed in plastic bags to prevent evaporation and placed in two large opened plastic containers (65 × 40 × 17 cm). The walls of the container were not transparent and each one received 18 Petri dishes. They were then placed outdoors where seeds received enough sunlight and daily average temperature ranged from 23 to 32 °C. The Petri dishes were left for 12 h in the plastic bags and opened for 30 min, allowing seeds or seedlings to be oxygenated. Tap water was added to keep moist during the experiment. Every morning, the Petri dishes were observed to monitor the number of germinated seeds. Records of daily temperature through the relevant experimental period were obtained from the meteorological office of Wuhan City.

In October (autumn), another germination experiment was done with the seeds from both groups stored in different conditions. These germination experiments and data recording was carried out following the same procedure as in July with a minor modification where the Petri dishes were not wrapped in plastic bags, because the temperature usually was below 30 °C during that period.

## Germination parameters

Two germination characteristics which are germination percentage (GP) and mean germination time (MGT) were estimated. MGT was determined according to the equation

of *Ellis & Roberts (1980)*: $MGT = \sum dn / \sum n$, where $n$ is the number of seeds newly germinated on days d, d refers as days counted from the beginning of germination test, and $\sum n$ is the total seeds germinated.

## Data analysis

A two level nested-ANOVA was performed for the data from the experiment conducted in a climate chamber to assess whether the difference in GP and MGT was significant between distribution range and populations within the range. Multiple comparisons by means of Tukey Contrasts was conducted to figure out which populations were significantly different from others in relation to MGT.

The results of one-way ANOVA tests indicated that there were generally no significant difference of GP and MGT among the populations within the same range, storage under the same condition and germinated in the same season (Table S1). Therefore, three-way ANOVA was used to determine significant differences in GP and MGT due to ranges, storage conditions and seasons on seed germination. To show the effect of every factor clearly, data were divided into groups according to the combination of two factors and $t$-tests were conducted to compare each couple defined by the third factor in different groups.

Before statistical analysis, the germination parameters (GP and MGT) were log-transformed to get distribution normality. Breusch-Pagan test was used to check homoscedasticity and Bonferonni outlier test was used for data –normality checking.

All statistical methods were performed using R software, version 3.2.1 (*R Core Team, 2015*).

# RESULTS

## Germination experiment in the climate chamber

The *S. vulgaris* seeds started to germinate at the 4th and 5th day after sowing for invasive and native populations, respectively. A high germination took place between the 4th and the 16th day (Fig. 1). After 19 days of germination experiment, all populations had > 80% of GP and 8 of the 12 populations had ≥ 90% GP (Fig. 2). The final GP (91.1%) was the same for invasive and native populations (Fig. 1). In addition, there was no significant difference in GP between the ranges and the populations within the ranges (two –level nested ANOVA, $df = 1$ and 10, $P > 0.05$).

The mean germination time (MGT) was not statistically different between the ranges, however, within the ranges, the populations were significantly different (two –level nested ANOVA, $df = 1$ and 10, for range: $F = 0.631$, for populations within range: $F = 2.398$, $P = 0.039$). The highest value of the MGT (13.51 days) was found in population from Tongjiang which belongs to the invasive range while the lowest value (seven days) was recorded in population from Oegstgeest that belongs to native range. Populations from Oegstgeest and Tongjiang were significantly different from each other, and these two populations significantly differ from the other 10 populations (Fig. 2).

## Outdoor germination experiments

Compared to the seeds germinated in summer, the GP of the seeds germinated in autumn was much higher, no matter which range the seeds from or under what kind of conditions
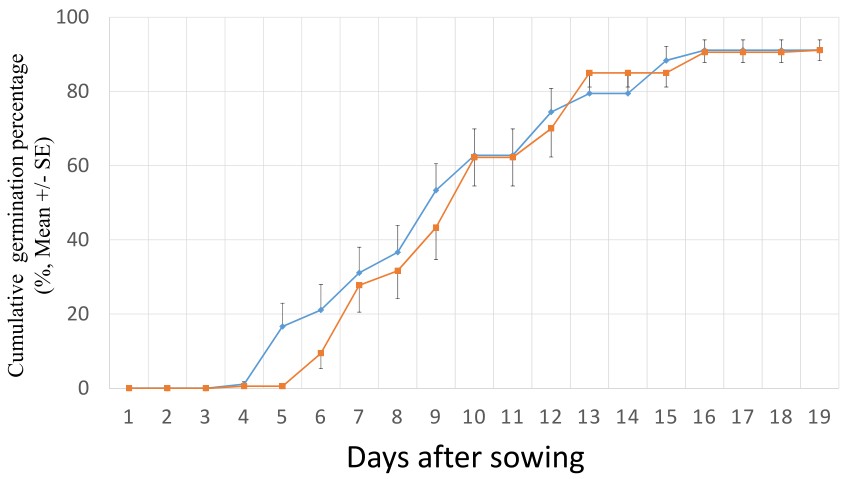

**Figure 1** **Cumulative germination percentage of *Senecio vulgaris* seeds from six native and six invasive populations in a climate chamber (15 °C, 12 h/12 h, dark/light) during 19 days.** ■ seeds from the invasive populations; ♦ seeds from the native populations. Ten seeds per each of three plants from each of the populations were used in this experiment.

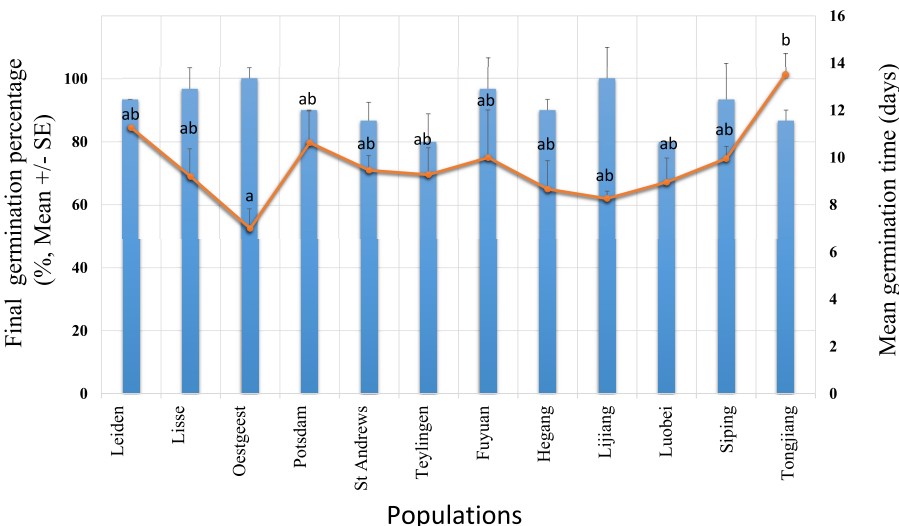

**Figure 2** **Final germination percentage (GP, bars) and mean germination time (MGT, dots) for *Senecio vulgaris* seeds from six native and six invasive populations in a climate chamber during 19 days.** The condition in the climate chamber was: 15 °C, 12 h/12 h, dark/light. Ten seeds per each of three plants from each of the populations were used in this experiment. No significant difference in GP was found between the ranges and the populations within the ranges. MGT was not statistically different between the ranges. Letters above the dots represent the results of multiple comparisons between populations in relation to MGT.

the seeds were stored (Figs. 3A and 4A). GP of *S. vulgaris* seeds were significantly different between the seasons (S), storage conditions (SC) and ranges (R). Only the interaction of SC × S was significant (Table 2). The influence of range and storage conditions on the GP depends on seasons.

In autumn, GP of the seeds from the invasive range was significantly higher than those from native range no matter what kind of conditions under which the seeds were stored. In

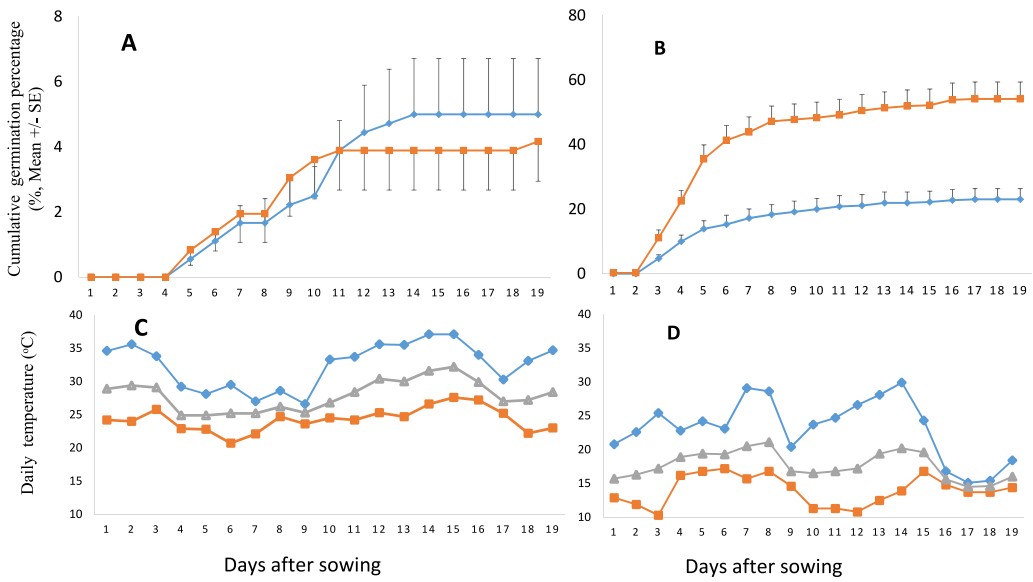

**Figure 3** **Cumulative germination percentage (GP) of *Senecio vulgaris* seeds and daily temperature during the experimental period.** (A–B) Cumulative GP of *Senecio vulgaris* seeds from six native and six invasive populations in an outdoor germination experiment during summer (A) and autumn (B) (♦ seeds stored at ambient room temperature, ca. 27 °C; ■ seeds stored at 4 °C ; seeds used in summer and autumn were stored for one month and four month, respectively). (C–D) Daily max (♦), min (■) and mean (▲) temperature during the experiment in summer (C) and in autumn (D). Data of temperature throughout the relevant experimental period were obtained from the meteorological office of Wuhan City.

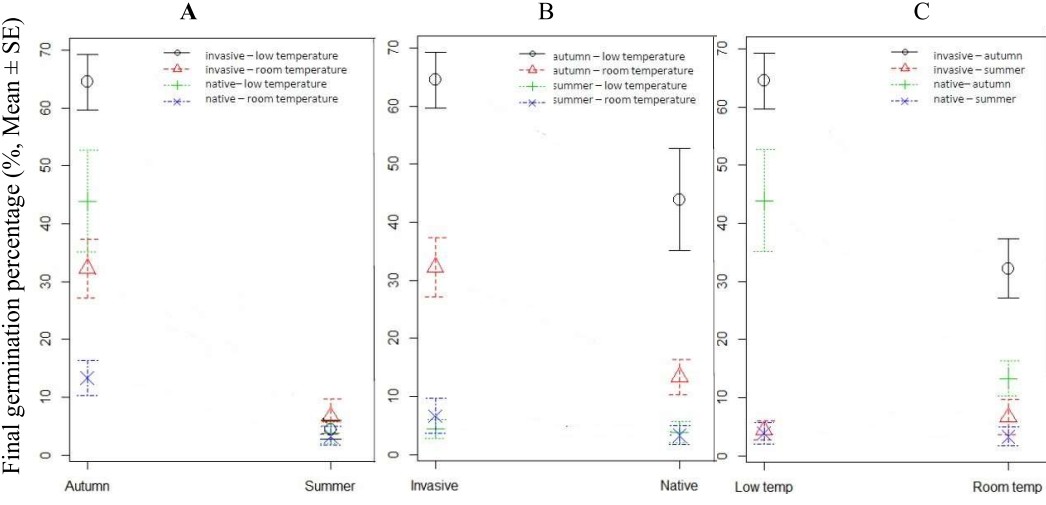

**Figure 4** **Comparison of the final germination percentage (GP) of *Senecio vulgaris* seeds germinated in different seasons, from different ranges and stored under different conditions.** (A) Comparison of final GP of seeds germinated in autumn and summer in 4 different groups divided according to the ranges and storage conditions (at 4 °C and ambient room temperature, ca. 27 °C); (B) Comparison of GP of seeds from native and invasive ranges in 4 different groups divided according to different storage conditions (at 4 °C and ambient room temperature, ca. 27 °C) and germinating seasons (summer and autumn); (C) Comparison of GP of seeds stored under different conditions in four different groups divided according to the ranges and germinating seasons.

**Table 2  Analysis of variance of final germination percentage (GP, %) and mean germination time (MGT, days) for *Senecio vulgaris* seeds from six native and six invasive populations.** Seeds were stored under different conditions (at 4 °C and ambient room temperature, ca. 27 °C) and germinated in different seasons (summer and autumn).

| Source of variation | | GP | MGT |
|---|---|---|---|
| | *df* | *F* | *F* |
| Range (R) | 1 | 10.0804[**] | 3.1723 |
| Storage condition (SC) | 1 | 6.7871[*] | 0.5177 |
| Season (S) | 1 | 117.2067[***] | 38.0943[***] |
| R × SC | 1 | 0.18 | 0.005 |
| R × S | 1 | 3.9866[*] | 0.0523 |
| SC × S | 1 | 6.3788[*] | 0.1391 |
| R × SC × S | 1 | 0.0033 | 0.0071 |

Notes.
[***]$P < 0.001$.
[**]$P < 0.01$.
[*]$P < 0.05$.

summer, there was no difference between invasive and native seeds in relation to GP (Fig. 4B). In autumn, final GP of seeds stored under 4 °C was 54.17%, and final GP of those stored under 27 °C was 22.78%, while the GP of the seeds germinated in summer was no more than 5% and was not different between the seeds stored under different conditions (Fig. 4C).

The *S. vulgaris* seeds started to germinate at the 2nd day after sowing in autumn, and in summer they started germination at the 5th day. Most germination in autumn took place between the 2nd and 8th day after sowing. MGT for the two seasons (summer and autumn) was statistically different and no interaction was revealed to be significant (Table 2). Higher MGT was recorded in autumn for both ranges and both storage conditions than in summer.

# DISCUSSION

## Do seasons and storage conditions have effect on seed germination of *S. vulgaris*?

Our result showed a high GP (91.51%) at constant temperature of 15 °C, indicating that the temperature conditions (15 °C) are appropriate or situated closer to the optimum germination temperature for *S. vulgaris* seeds. In summer, the GP was very low (4.5%), compared to that of autumn (38.6%). This could be due to the high temperature during the experimental period in summer when the average temperature ranged from 25 to 30 °C, and the maximum temperature ranged from 27 to 37 °C (Fig. 3C). Our results agree with previous studies that reported the optimum growing or germination temperature for *S. vulgaris* ranging between 10 °C and 25 °C, above or beyond these limits, the GP declines (*Popay & Roberts, 1970a*; *Ren & Abbott, 1991*; *Walkey & Cooper, 1976*).

In autumn, seeds stored at 4 °C displayed higher GP than those stored at ambient room temperature (Fig. 4C). The reason for this might be that seeds stored at room temperature
from July to October were subjected to the variation in temperature (Fig. S1) and humidity that is high in summer, and resulting in a loss of viability, or secondary dormancy. *Popay & Roberts (1970b.)* found that dry *S. vulgaris* seeds at high temperature 35 °C for 10 weeks got high GP at germination seeds, but some previous studies showed that storage at room temperature often resulted in low seed germination (*Nasreen, Khan & Mohmad, 2000*). Hence, it is interesting to confirm what is the real reason by further experiments.

### Do GP and MGT of *S. vulgaris* seeds vary depending on their origination?

In climate chamber at 15 °C, the MGT was different between populations, implying that the origin might influence the speed of seed germination. However, there was no difference in GP between the ranges, or between populations within the range. Under this optimum germination condition and after a rather long period (19 days), every *S. vulgaris* seed with good quality could germinate. This might be the reason why we did not found different final GP between populations and ranges.

In the outdoor experiment during autumn, the GP was statistically different between the ranges (Table 2, Fig. 4B). We also found, in autumn, the seeds stored at 4 °C and from invasive plants gained about 65% GP, those from native plants gained about 45%; and the difference of GP between the range was significant (Table 2, Fig. 4B). The high germination in invasive populations may be due to their ability to thrive in the new environment, probably based on the rapid adaptation or their genotype evolution as it was reported for many species (*Hierro et al., 2009*; *Skálová, Moravcová & Pyšek, 2011*; *Leiblein-Wild, Kaviani & Tackenberg, 2014*).

Additionally, *S. vulgaris* has a wide distribution range (*Holm et al., 1979*) which means that the geographical variation may lead to the difference in germination behavior as it was detected in other species (*Lindauer & Quinn, 1972*; *Thompson, 1975*). Our results are consistent with previous studies carried out with *S. vulgaris* seeds collected from different areas such as Scotland and South Spain (*Ren & Abbott, 1991*); Kentucky and Michigan (*Figueroa et al., 2010*); Scotland and Yugoslavia (*Richards, 1975*). The present results showed that the germination behavior occurred differently according to the geographic origin of the seeds.

### CONCLUSIONS

We observed that seeds of *S. vulgaris* could germinated in short time after sowing and had high GP at 15 °C in a climate chamber and in an outdoor germination experiment in autumn. Low GP was observed for seeds germinated in summer and seeds kept at ambient room temperature (about 27 °C). This indicated that high temperature in summer has negative impact on the germination and might cause viability lost or secondary dormancy to of *S. vulgaris* seeds. The present results showed that the germination behavior occurred differently according to the geographic origin of the seeds. We found that seeds from invasive plants gained high GP than those from native ones in the outdoor experiment during autumn. This indicated that the invasive *S. vulgaris* could be more adapted to some environments in China. Our study offers a clue to explore which factor limits the

distribution of *S. vulgaris* in some regions of China by explaining why, in the cities in South-East China and central China such as Wuhan, *S. vulgaris* cannot establish natural and viable populations.

## ACKNOWLEDGEMENTS

Colleagues in the School of Environmental Studies at China University of Geosciences (Wuhan) and Institution of Biology in Leiden University are thanked for helping in seed collection. We further acknowledge Harold W.T. Mapoma, Prosper Laari and Tananga M. Nyirenda for their valuable comments on the manuscript. The two anonymous reviewers should receive our warmest gratitude for their invaluable comments that significantly improved this paper.

### Funding

This work is supported by the Fundamental Research Funds for the Central Universities (CUG 130411) and National Natural Science Foundation of China (31570537 and 31200425) granted to Dandan Cheng. Viet Thang Nguyen and Noel Ndihokubwayo are supported by the Chinese Scholarship Council (CSC) for the study in China. The funders had no role in study design, data collection and analysis, decision to publish, or preparation of the manuscript.

### Grant Disclosures

The following grant information was disclosed by the authors:
Fundamental Research Funds for the Central Universities: CUG 130411.
National Natural Science Foundation of China: 31570537, 31200425.
Chinese Scholarship Council (CSC).

### Competing Interests

The authors declare there are no competing interests.

### Author Contributions

- Noel Ndihokubwayo conceived and designed the experiments, performed the experiments, analyzed the data, wrote the paper, prepared figures and/or tables, reviewed drafts of the paper.
- Viet-Thang Nguyen performed the experiments, reviewed drafts of the paper.
- Dandan Cheng conceived and designed the experiments, analyzed the data, contributed reagents/materials/analysis tools, wrote the paper, prepared figures and/or tables, reviewed drafts of the paper.

### Data Availability

The raw data has been supplied as Data S1.

**PeerJ** ________________________________________________

## Supplemental Information

Supplemental information for this article can be found online at http://dx.doi.org/10.7717/peerj.2346#supplemental-information.

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
