# Peer review of "Effects of origin, seasons and storage under different temperatures on germination of Senecio vulgaris (Asteraceae) seeds"

_PeerJ, doi:10.7717/peerj.2346_

## Round 0.1 · original submission · Major Revisions

· Academic Editor

Major Revisions

Both reviews were quite critical, especially considering the description of the experimental design. Please, improve the manuscript according to the suggestions of the reviewers and note the attached annotated manuscript from Reviewer 2. An improvement to the English language is also necessary.

Reviewer 1 ·

Basic reporting

(1) Unfortunately, the professional English used in this paper is sub par. There are many grammatical errors and some phrasing is very confusing. I suggest that this paper be revised by someone whose first language is English to clear up these issues.

(2) The literature cited seemed well-referenced and very relevant.

(3) The structure of the paper does conform to the PeerJ standards.

(4) The figures are well-done and easy to follow and understand.

(5) Raw data is included.

Experimental design

I have serious concerns about the experimental design utilized in this study.

(1) First of all, it appears that replicates were not used. As I understand what is written, only one unit of seeds per treatment was used, thus, no replicates. IF replicates were used, this is not clear from the description provided. In these types of seed studies, it is essential, for proper statistical analysis, to use replicates (for example, three Petri dishes of seeds from Leiden). IF replicates were not used, the study needs to be conducted again. IF replicates WERE used, then the language of the paper needs to be altered to make this fact clear.

(2) Also, the numbers of seeds used were too low. For example, using only 10 seeds per treatment would not account for the often large variation in germination patterns observed in many species. I understand that it might be difficult to obtain large numbers of seeds, but 10 per treatment is just too few.

Validity of the findings

(1) The findings were very interesting and are possibly very significant. However, the apparent errors in the methodology undermine potentially significant results.

(2) The conclusions on dormancy states do not appear to be entirely correct. The claim is made that one subspecies is very dormant. In all treatments, however, seed germination begins relatively soon after imbibition (of water) and final germination percentages are relatively high. This does not indicate the presence of dormancy.

It seems to me that the lack of germination is due to exposure to temperatures not conducive to germination. If all seeds were placed under appropriate temperature regimes, germination would occur in most viable seeds. Thus, seeds not germinating is due to the lack of adequate environmental conditions (i.e., temperature) and not due to the presence of dormancy. Seeds germinated in one season but not the other. This could simply be due to lack of appropriate germination temperatures and not the presence of dormancy.

I do realize that some populations of S. vulgaris do produce dormant seeds. I just do not think the case is made here for dormancy. The study needs to be improved so as to rule out temperature as the inhibiting factor and to show that the seeds truly exhibit dormancy.

Additional comments

The reasons for the study and the questions asked are very good and relevant. The proposed questions should make for a very good study with potentially significant results. That stated, the apparent errors in methodology present some serious concerns about making appropriate conclusions. Furthermore, some conclusions themselves seem unlikely. The methods and conclusions need to be greatly revised before this paper is suitable for publication.

Reviewer 2 ·

Basic reporting

The manuscript need English editiom. Information is original, but, it has many flaws.

Experimental design

I could not evaluate the experimental design because I did not understand which was the experimental unit, I understood that all the treatments were placed twice, but this do not implicate that replicate the experimental unit be strictly necessary.

Validity of the findings

I could not evaluate the experimental design because I did not understand which was the experimental unit, I understood that all the treatments were placed twice, but this do not implicate that replicate the experimental unit be strictly necessary.

Additional comments

The topic of the manuscript is interesting, but it has many flaws. I did comments on the PDF, this could help authors to prepare other manuscript or experimental design. Temperatures in the room or laboratory ought to be documented correctly, outdoor and indoor temperatures are different and they can also vary depending on the microsite where they were taken. Results of the post hoc statistical tests ought to be shown on figures, Discussion contain more published information than a real discussion of the obtained results. Thus elucubration is common along this section. Methods ought to be the most clear possible in order to determine the validity and relevance of the research

Annotated reviews are not available for download in order to protect the identity of reviewers who chose to remain anonymous.

---

## Round 0.2 · Major Revisions

· Academic Editor

Major Revisions

The revised manuscript has improved, but there are still concerns by the reviewers, which you should try to address. Please, note the attached annotated manuscript.

Further corrections:

line 22: Replace 'originated' with 'originating'
line 33: Replace 'lost' with 'loss'
line 91: Rephrase to: The first germination experiment which used....

Reviewer 1 ·

Basic reporting

No Comments

Experimental design

I still have one problem with the experimental design: I do NOT think enough seeds were used per treatment. Only ten seeds were used per replicate. I do not think that this provides a true representation of dormancy continuum states in a population.

Validity of the findings

No Comments

Additional comments

Also, the authors refer to seeds from different "families". Is the term "family" being used to represent different species or genera? The only family represented here is the Asteraceae. So, I'm not sure why there is a reference to populations from different "families". Do you mean something else when you use this term?

Reviewer 2 ·

Basic reporting

The manuscript contain interesting information but it requires important changes

Experimental design

Please see the PDF

Validity of the findings

Findings are important but it is necessary many changes in the statistical analysis

Additional comments

Dear editor:
I am concern because the manuscript is not ready for publication. I placed comments on the PDF. An additional comment: The manuscript is wordy and extremely repetitive. I suggest a general section with the general procedures. The most important is that the statistical analysis ought to be modified in the sense that is indicated on the PDF. I suggest eliminate the maternal families, in the context of the results is irrelevant. This is the main problem in this manuscript. Figures ought completed with the statistical information that I indicated on the PDF. After that it is possible resubmit in Peer J or in other Journal.

Annotated reviews are not available for download in order to protect the identity of reviewers who chose to remain anonymous.

---

## Round 0.3 · Minor Revisions

· Academic Editor

Minor Revisions

The manuscript is close to acceptance now. There is still the necessity for some language editions and formal improvements:
Line 27: replace ‘effect’ with ‘effects’
Line 34: replace ‘what’ with ‘which’
Line 36: replace ‘can’t’ with ‘cannot’
Line 43: replace ‘that’ with ‘which’
Line 43: change to: …..and is….
Line 44: Replace ‘wildly’ with ‘widely’
Line 52: change to: …dispersion. Plants from…..
Line 237: change to:….according to the geographic origin of the seeds.
Line 240: replace ‘environment’ with ‘environments’
Line 241: replace ‘what’ with ‘which’
Line 242: replace ‘can’t’ with ‘cannot’
Line 297: replace ‘Austria..’ with ‘Austria.’

Table 1: Check formation: ‘Time’ is different font. Check line 8 within table.
Table 2: Last line is not clear: Significance codes: ***P<0.001; **P<0.01; *P<0.05.
Figure legend 1: Last sentence: Ten seeds……..families…..
Figure legend 2: 4th line: replace ‘dark/light .’ with ‘dark/light.’
Same line: Ten seeds….
Figure legend 3: replace ‘experiment period’ with ‘experimental period’
Replace: ‘a outdoor’ with ‘an outdoor’
Figure legend 4: 4th line: …divided according to….
Last line: … divided into….
After complying these changes the manuscript can be accepted for publication.

---

## Round 0.4 · accepted · Accept

· Academic Editor

Accept

The authors complied all suggested changes for improvement of the manuscript, which can be accepted now.

Congratulations to the authors!